# Improvement of Colonoscopic Image Quality Using a New LED Endoscopic System with Specialized Noise Reduction

**DOI:** 10.3390/diagnostics15121569

**Published:** 2025-06-19

**Authors:** Naohisa Yoshida, Masahiro Okada, Yoshikazu Hayashi, Reo Kobayashi, Ken Inoue, Osamu Dohi, Yoshito Itoh, Ryohei Hirose, Lucas Cardoso, Kohei Suzuki, Tomonori Yano, Hironori Yamamoto

**Affiliations:** 1Department of Molecular Gastroenterology and Hepatology, Graduate School of Medical Science, Kyoto Prefectural University of Medicine, Kyoto 602-8566, Japan; reipanna@koto.kpu-m.ac.jp (R.K.); keninoue71@koto.kpu-m.ac.jp (K.I.); osamu-d@koto.kpu-m.ac.jp (O.D.); yitoh@koto.kpu-m.ac.jp (Y.I.); 2Department of Medicine, Division of Gastroenterology, Jichi Medical University, Tochigi 329-0498, Japan; r0811mo@jichi.ac.jp (M.O.); hayashi@jichi.ac.jp (Y.H.); m14065ks@jichi.ac.jp (K.S.); tomonori@jichi.ac.jp (T.Y.); yamamoto@jichi.ac.jp (H.Y.); 3Department of Infectious Disease, Graduate School of Medical Science, Kyoto Prefectural University of Medicine, Kyoto 602-8566, Japan; ryo-hiro@koto.kpu-m.ac.jp; 4Department of Colorectal Surgery, Hospital Felício Rocho, Belo Horizonte 30110-934, MG, Brazil; cardluc@hotmail.com

**Keywords:** linked color imaging, blue light imaging, LED, colonoscopy

## Abstract

**Background/Objectives:** A new LED endoscopy system featuring advanced noise-reduction technology, the EP-8000 with the EC-860ZP colonoscope (Fujifilm), was introduced in 2024. We evaluated the improvements in colonoscopic image quality of this system, comparing it with a previous system/scope (VP-7000/EC-760ZP). **Methods:** This is a multicenter, observational study. From January 2024 to February 2025, 150 patients undergoing colonoscopy at two institutions were enrolled. Images of the cecum and lesions were captured using white light imaging (WLI), blue light imaging (BLI), and linked color imaging (LCI) under similar conditions. Participants were divided into three groups: Group 1 (EP-8000+EC-860ZP; 50 cases), Group 2 (EP-8000+EC-760ZP; 50 cases), and Group 3 (VP-7000+EC-760ZP; 50 cases). Cecal and lesion images were evaluated for brightness, halation, and visibility using a four-point scale (1 = poor to 4 = excellent) by endoscopists and original values by image-analysis software. **Results:** In cecal images, the endoscopists’ scores in Group 1 were significantly better than in Group 3 for brightness (WLI: 3.71 ± 0.55 vs. 3.51 ± 0.58, *p* < 0.001, BLI: 3.15 ± 0.85 vs. 2.23 ± 0.92, *p* < 0.001; LCI: 3.83 ± 0.42 vs. 3.54 ± 0.58, *p* < 0.001) and for halation (WLI: 3.60 ± 0.51 vs. 3.18 ± 0.59, *p* < 0.001, BLI: 2.99 ± 0.69 vs. 2.71 ± 0.78, *p* < 0.001; LCI: 3.33 ± 0.60 vs. 3.10 ± 0.58, *p* < 0.001). Software analysis confirmed that Group 1 had superior brightness values compared with Group 3 for WLI, BLI, and LCI, as well as lower halation values for WLI and LCI. Regarding lesion images, brightness, halation, and visibility for WLI, BLI, and LCI were superior in Group 1 than in Group 3. **Conclusions:** The new LED system provided improvements in brightness, halation, and lesion visibility.

## 1. Introduction

A laser endoscopic system, LASEREO (VP-4450HD, Fujifilm, Tokyo, Japan), was introduced in 2012, employing two-wavelength lasers (410 nm and 450 nm) for white light imaging (WLI) and blue laser imaging (BLI), which enhances lesion characterization and detection [1,2,3,4,5,6]. The bandwidths of the two lasers were less than about 2 nm. The laser with a 450 nm wavelength made phosphor irradiate an illumination similar to that of a xenon lamp, which was used in the endoscopic system regularly. BLI is a narrow-band observation mode that has two modes—regular BLI mode for lesion characterization and BLI-bright mode with high illumination for lesion detection. However, the BLI can appear dark, and residual fluids may become reddish under poor preparation. Linked color imaging (LCI) was subsequently developed as a kind of narrow-band light observation mode, providing a brighter field of view and turning the fluid yellowish using two lasers, thereby making neoplastic lesions and inflammatory areas appear reddish [7,8]. Multiple studies have demonstrated improved lesion visibility and polyp detection using LCI [9,10,11,12,13,14,15]. An LED-based endoscopic system, ELUXEO (VP-7000, Fujifilm) with a dedicated scope (EC-760ZP), was later introduced in 2017 in Western countries and in 2020 in Japan [16,17,18], utilizing four-color LEDs (blue-violet, blue, green, red). Multi-light technology provides high-quality images with blue light imaging (BLI) as well as LCI for narrow-band light observation. Many randomized controlled trials reported that LCI reduces polyp miss rates and improves detection compared with WLI [19,20,21,22,23]. A systematic review of LED and laser colonoscopes similarly showed higher adenoma and polyp detection rates for LCI over WLI [24]. In 2024, a new LED system, EP-8000 (Fujifilm), featuring Triple Noise Reduction (TNR) and Extended Dynamic Range Image Processing (EDRIP), was launched, delivering low-noise, bright images with minimal halation. The present study aims to evaluate the image quality improvements of this latest LED system in colonoscopy.

## 2. Materials and Methods

This observational, multicenter, prospective, and retrospective study involves patients undergoing colonoscopy at Kyoto Prefectural University of Medicine and Jichi Medical University from January 2024 to February 2025. Eligible patients underwent total colonoscopy for one of the following indications: (1) detailed examination of various symptoms (e.g., abdominal pain, constipation, anemia, hematochezia); (2) surveillance after polyp or cancer resection; (3) screening; and (4) positive fecal occult blood test. Sessile serrated lesions (SSL), low-grade adenomas, high-grade dysplasia, and T1a cancers (diagnosed endoscopically and confirmed histologically) were included. Exclusion criteria were: T1b–T4 colorectal cancer, inflammatory bowel syndrome, hereditary polyposis syndromes, and prior surgery involving the cecum. Colonoscopies were performed by two experts to ensure high-quality imaging.

Cecal and lesion images were captured under identical conditions (angle, distance, insufflation volume) using WLI, BLI, and LCI (Figure 1). The endoscopic systems and scopes were categorized as follows: Group 1: EP-8000+EC-860ZP, Group 2: EP-8000+EC-760ZP, and Group 3: VP-7000+EC-760ZP. During the study period, 150 cases were prospectively collected into each group. Two endoscopists (N.Y. and M.O.) reviewed the images and excluded lesions where conditions were not comparable (e.g., air insufflation, angle, residual fluid, distance). The final number of cases was 50 (52 lesions) in Group 1, 50 (62 lesions) in Group 2, and 50 (56 lesions) in Group 3. Four endoscopists (one expert and one non-expert from each institution) independently assessed the images. They were blinded to the lighting modality and lesion characteristics. According to our previous paper, experts were those who had performed ≥5000 colonoscopies and ≥50 colonoscopies with LCI observation [25]. Non-experts had less experience either in total colonoscopy volume or in LCI use.

## 3. Outcomes

The primary outcome was the comparison of brightness and halation among the novel LED system (EP-8000) and the previous system (VP-7000) for cecal and lesion images (WLI, BLI, and LCI), as evaluated by endoscopists using a 4-point scale (1 = poor, 2 = fair, 3 = good, 4 = excellent) (Appendix A). The brightness score was defined as follows: 4, indicating excellent clarity for both close and distant views; 3, indicating good clarity for both close and distant views; 2, indicating fair clarity for close views but unclear for distant views; and 1, indicating unclear for both close and distant views. The halation score was defined as follows: 4, indicating no halation; 3, indicating halation affecting less than 5% of the endoscopic view; 2, indicating halation affecting 5–19% of the endoscopic view; and 1, indicating halation affecting 20% or more of the endoscopic view. Additionally, an objective comparison of brightness and halation in the cecal images of WLI, BLI, and LCI among 15 cases was performed, showing excellent bowel preparation without any residual liquid in each group, using specialized image-analysis software (Figure 2 and Figure 3). Regarding brightness, its average was calculated using the standard formula based on JPEG image RGB values: Brightness = 0.299 × R + 0.587 × G + 0.114 × B, which produces a value in the range of 0–255 (black to white) according to our previous paper [13]. Regarding halation, its ratio was originally defined as areas with brightness levels of 90% or higher, and the halation ratio was defined as the proportion of the halation region area to the total endoscopic imaging area. The halation ratio was calculated using the formula: Halation ratio = Halation region area ÷ Endoscopic imaging area. Regarding software analysis, no image enhancement or alteration of image appearance was performed. The integrity of the endoscopic findings was fully preserved, and all original image data were retained.

The secondary outcomes were the visibility of lesion images (WLI, BLI, LCI) and BLI/LCI images with magnification, graded on a 4-point polyp-visibility scale (1 = poor, 2 = fair, 3 = good, 4 = excellent). According to previous reports, the visibility score was defined as follows: score 4 indicating excellent visibility (easy to detect); 3 indicating good visibility (if an endoscopist looked in the direction of the polyp on the monitor, it would be easy to detect); 2 indicating fair visibility (where it would be difficult to detect the polyp without careful observation); and 1 indicating poor visibility (Figure 4, Appendix A) [5,8,13]. Assessments of endoscopist experience level (expert vs. non-expert) regarding the scoring consistency and diagnostic accuracy based on the Japan narrow-band imaging (NBI) Expert Team (JNET) classification were analyzed [26].

## 4. LED Endoscopes

The novel LED endoscopic system and magnifying colonoscope used in this study were the EP-8000 and EC-860ZP (Fujifilm Co., Tokyo, Japan). This system utilizes different four-color LEDs (blue-violet, blue, green, amber-red) compared with the previous system and incorporates two key features: TNR and E-DRIP (Figure 5).

TNR incorporates three noise suppression technologies, each derived from different imaging modalities, to reduce noise in blood vessels and mucosal surface structures, thereby improving visibility compared with conventional systems. These technologies include: (1) conventional endoscopic noise suppression, which removes sudden noise by comparing consecutive frames. (2) X-ray-based noise suppression, which enhances the clarity of thin blood vessels. (3) Ultrasound-based noise suppression, which improves the contrast of mucosal details.

E-DRIP optimizes tone control, brightens dark areas, and expands the dynamic range. Additionally, its improved exposure control minimizes the occurrence of “crushed blacks” and “clipped highlights”. These advancements result in brighter images with reduced halation and enhanced visibility even at a distance (Appendix A).

This system enables us to perform WLI, BLI, and LCI and also has a new observational mode named amber-red color imaging for enhancing vessels during endoscopic submucosal dissection.

The previous LED endoscopic system used in this study consisted of a BL-7000 light source, a VP-7000 processor, and an EC-760ZP-V/M endoscope (Fujifilm Co., Tokyo, Japan). After consultation between the two institutions involved, the image settings for the novel LED system were chosen as follows: WLI: A8 (without red enhancement), BLI: A6, C2, LCI: A5, C1. These settings aimed to achieve high-contrast images with minimal noise. By contrast, the previous system’s settings were H, +4 (Detail Hi), and +4 (Detail Low) for WLI, B8/C2 for BLI, and B8/C3 for LCI, in accordance with a previous report [25].

Bowel preparation was assessed using the Aronchick score [27]. Scores of “Excellent” and “Good” were classified as “Good”, while “Fair” and “Poor” were classified as “Bad”. Polyp size was determined by measuring the maximum diameter, based on resected specimens, snares, or biopsy forceps. Polyps were categorized as either polypoid or non-polypoid according to the Paris classification [28]. Histopathological diagnosis followed the World Health Organization (WHO) classification and the Japanese Society for Cancer of the Colon and Rectum guidelines [29,30]. Sessile serrated lesions (SSL) were defined according to the WHO criteria. High-grade adenoma and intramucosal carcinoma were grouped as high-grade dysplasia, and T1a lesions were defined in accordance with Japanese guidelines.

This study was conducted in compliance with the Declaration of Helsinki. The Institutional Review Boards of Kyoto Prefectural University of Medicine and Jichi Medical University approved the protocol (ERB-C-3227), and an opt-out procedure for patient consent was implemented at both institutions.

## 5. Statistical Analysis

Continuous variables (e.g., age and lesion size) were presented as mean ± standard deviation (SD). Categorical variables were compared using the chi-squared test with Yates’ correction, whereas continuous variables were analyzed using the Mann–Whitney U test. All statistical tests were two-tailed, and *p*-values < 0.05 were considered statistically significant. Statistical analyses were performed using SPSS software (version 22.0; IBM Japan, Ltd., Tokyo, Japan).

## 6. Results

In Groups 1, 2, and 3, the mean ± SD patient ages were 68.2 ± 10.4, 66.0 ± 14.4, and 66.8 ± 11.6 years, respectively (*p* = 0.706) (Table 1). The rates of good bowel preparation were 82.0%, 78.0%, and 86.0% (*p* = 0.581). Lesion sizes (mean ± SD) were 4.1 ± 3.5, 4.1 ± 3.2, and 4.5 ± 3.3 mm (*p* = 0.753).

Regarding cecal images, there were significant differences in the endoscopists’ evaluations of brightness between Groups 1 and 3 for WLI (3.71 ± 0.55 vs. 3.51 ± 0.58, *p* < 0.001), BLI (3.15 ± 0.85 vs. 2.23 ± 0.92, *p* < 0.001), and LCI (3.83 ± 0.42 vs. 3.54 ± 0.58, *p* < 0.001) (Figure 6, Appendix A). As for halation, Groups 1 and 3 differed significantly for WLI (*p* < 0.001), BLI (*p* < 0.001), and LCI (*p* < 0.001).

Regarding the software-based evaluation of cecal images, significant differences in brightness were observed between Groups 1 and 3 for WLI (122.7 ± 6.6 vs. 110.7 ± 8.7, *p* < 0.001), BLI (99.1 ± 10.9 vs. 69.3 ± 11.5, *p* < 0.001), and LCI (139.4 ± 9.2 vs. 113.1 ± 10.7, *p* < 0.001) (Figure 6, Appendix A). As for halation, Group 1 and Group 3 differed significantly not for BLI (*p* = 0.173), but for WLI (*p* = 0.031) and LCI (*p* = 0.010) (Figure 7, Appendix A).

In the endoscopists’ evaluations of lesion images, brightness scores differed significantly between Groups 1 and 3 for WLI (3.72 ± 0.50 vs. 3.51 ± 0.57, *p* < 0.001), BLI (3.51 ± 0.67 vs. 3.01 ± 0.71, *p* < 0.001), and LCI (3.89 ± 0.32 vs. 3.60 ± 0.56, *p* < 0.001) (Table 2). For halation, Groups 1 and 3 differed significantly for WLI (*p* < 0.001), BLI (*p* < 0.001), and LCI (*p* < 0.001). Visibility scores were significantly higher in Group 1 than in Group 3 for WLI (*p* = 0.005), BLI (*p* < 0.001), and LCI (*p* < 0.001). Additionally, those in non-polypoid lesions were significantly higher in Group 1 than in Group 3 for WLI (*p* = 0.016), BLI (*p* < 0.001), and LCI (*p* = 0.002) and those in polypoid lesions were significantly higher in Group 1 than in Group 3 for BLI (*p* < 0.001) and LCI (*p* < 0.001) (Appendix A).

Regarding images with magnification, the brightness of lesion images differed significantly between Groups 1 and 3 for BLI (*p* < 0.001) and LCI (*p* < 0.001) and between Groups 2 and 3 for BLI (*p* < 0.001) and LCI (*p* = 0.034) (Appendix A). Halation scores in Groups 1 and 3 were significantly different for BLI (*p* < 0.001) and LCI (*p* = 0.005), and in Groups 2 and 3 for BLI (*p* < 0.001) and LCI (*p* = 0.027). Visibility also differed significantly between Groups 1 and 3 for BLI (*p* < 0.001) and LCI (*p* < 0.001).

Diagnostic accuracy was 91.5% in Group 3 vs. 84.6% in Group 1 (*p* = 0.026) (Appendix A).

## 7. Discussion

In the current study, we demonstrated that the novel system and scope improved the brightness and halation of WLI, BLI, and LCI compared with the previous system and scope, as evidenced by both subjective endoscopist evaluations and objective image software assessments. To the best of our knowledge, this is the first study to show the efficacy of the novel LED endoscopy system, EP-8000. These improvements are likely achieved through two special features, such as TNR and E-DRIP, that reduce noise and enhance brightness. Additionally, we developed a software-based method for evaluating halation, which we believe can be beneficial for future endoscopy development.

In this study, we evaluated cecal images using both endoscopists and software. Software-detected improvements in WLI, BLI, and LCI were consistent with endoscopist evaluations in terms of brightness. Regarding halation, software analysis showed significant improvements in WLI and LCI with the new system, aligning with endoscopist evaluations except for BLI. Based on these findings, software assessment may serve as an objective tool for examining new endoscopic systems without a subjective endoscopist’s input. However, to enhance consistency, the halation threshold in software assessments should be refined in future studies.

Regarding lesion visibility scores, our previous multicenter study compared the previous LED endoscopy system (VP-7000+BL7000) and a laser endoscopic system (VP-7000+LL-7000). In the LED system, multi-light technology with four LED lights enables WLI, BLI, and LCI modes. On the other hand, in the laser system, two lasers and phosphor enable WLI, BLI and LCI modes to perform. In the study, we assessed the visibility of 63 non-polypoid lesions using ratings by 12 endoscopists (six experts and six non-experts) [13]. The mean ± SD polyp size in the study was 24.5 ± 13.4 mm. All lesions were observed by both laser and LED endoscope and non-inferiority in polyp visibility was observed for both LCI (3.35 ± 0.86 vs. 3.36 ± 0.85, *p* < 0.001) and WLI (3.05 ± 0.98 vs. 3.08 ± 0.96, *p* < 0.001) [25]. Irrespective of the endoscopist’s experience level, lesion location, size, or histopathology, the visibility scores for LCI and WLI in laser endoscopy were non-inferior to those in LED endoscopy. In the current study, visibility scores are significantly higher in Group 1 (novel system) than in Group 3 (previous system) for WLI (3.28 ± 0.73 vs. 3.09 ± 0.82, *p* = 0.005), BLI (3.69 ± 0.54 vs. 3.36 ± 0.69, *p* < 0.001), and LCI (3.63 ± 0.59 vs. 3.31 ± 0.71, *p* < 0.001). However, the mean lesion sizes in the three groups (4.1–4.5 mm) are smaller than those in the aforementioned studies, which may have influenced the visibility scores. Additionally, lesions were different in each group with small numbers. Validation studies with large numbers will be expected to clarify the difference.

There are several specialized observational modes with different mechanisms for improving lesion detection, in addition to BLI and LCI. In NBI, optical filters that allow narrow-band light at wavelengths of 415 and 540 nm to pass through are mechanically inserted between a xenon lamp and an RGB rotary filter [31]. NBI enhances vascular patterns and pit-like structures and is used for both lesion detection and characterization. Similarly to BLI, residual fluid appears reddish under NBI. Texture and Color Enhancement Imaging (TXI) is another imaging mode developed to improve lesion visibility by enhancing three image-related factors: texture, brightness, and color [32]. The EVIS X1 (CV-1500; Olympus Co., Tokyo, Japan) is an endoscopic LED system that uses five LED light sources, enabling the use of TXI as well as improved brightness and image resolution in NBI. In TXI, residual fluid appears yellowish, and the endoscopic view remains bright even under poor bowel preparation. Several randomized controlled trials (RCTs) have demonstrated the efficacy of NBI and TXI in polyp detection compared with WLI [33,34,35,36].

For analyzing these different observational modes, the polyp visibility score used in the present and previous studies is a kind of subjective measure. Nevertheless, several investigations have used polyp visibility scores to compare various endoscopic observation methods. A previous study evaluated NBI and TXI against WLI based on polyp visibility scores and found that NBI and TXI provided higher visibility for serrated colorectal polyps, including SSLs and adenomas [37,38]. Moreover, in previous studies, LCI showed better scores than WLI in videos and still images of lesions [8,13]. Our previous study included an objective color difference (CD) analysis of WLI and LCI, calculating the difference between the lesion and surrounding mucosa in eight directions, which showed a strong correlation between CD values and polyp visibility scores for both WLI and LCI [15]. These findings suggest that, although polyp visibility scoring is subjective, it is a useful tool for assessing different observation modes.

Many RCTs have supported the benefits of LCI. For instance, a European RCT of LED colonoscopy showed a significantly lower adenoma miss rate in the right-sided colon (cecum and ascending colon) with LCI compared with WLI (11.8% vs. 30.6%, *p* < 0.001) [19]. A Brazilian study using laser colonoscopes showed a higher ADR with LCI than with WLI (56.9% vs. 43.2%, *p* = 0.03) [12]. However, two studies presented somewhat different results. An RCT regarding the LASER colonoscope study reported that LCI did not show superiority throughout the entire colon but did in the descending and sigmoid colon [23]. Another RCT for LED colonoscopes found LCI to be superior regardless of location, and an endoscopic cap further enhanced its efficacy possibly because of the higher brightness of the LED colonoscope [39]. Our current study indicates that the novel LED colonoscope improves the visibility of WLI, BLI, and LCI more than the previous LED colonoscope. As a result, the novel LED system could potentially increase ADR more effectively than the previous system, although further investigation is warranted.

According to our results, the clinical use of the new LED system—with enhanced brightness and reduced halation—is effective for both routine and magnified observation using WLI, BLI, and LCI. In cases of poor bowel preparation with substantial residual fluid, the increased brightness of WLI and LCI may facilitate improved lesion detection. Furthermore, the brighter BLI provided by the new system enables its use for lesion detection, overcoming the limitation of insufficient brightness seen in previous BLI systems. Further analysis is warranted to validate and clarify these findings.

There were several limitations in the current study. First, each group had a relatively small number of lesions. Second, we focused solely on overall brightness, halation, and lesion visibility in the cecum and did not analyze lesion location, size, histopathology, or bowel preparation status in depth. While improved polyp visibility scores may correlate with better polyp detection, actual detection rates in clinical practice can be influenced by many additional factors. Furthermore, only Japanese endoscopistsparticipated in this evaluation. Therefore, an international assessment is needed for global efficacy.

## 8. Conclusions

Our multicenter study demonstrated that the new LED endoscopy system significantly improves brightness, halation, and visibility of cecal locations and lesion images under endoscopist and software assessments. The software’s objective assessment was consistent with the endoscopists’ subjective assessment and can be used for the development and evaluation of new endoscopic systems.

## Figures and Tables

**Figure 1 diagnostics-15-01569-f001:**
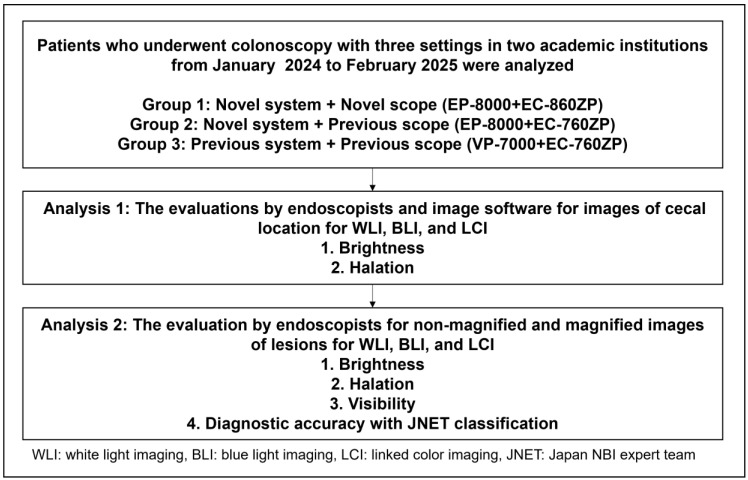
A study flow.

**Figure 2 diagnostics-15-01569-f002:**
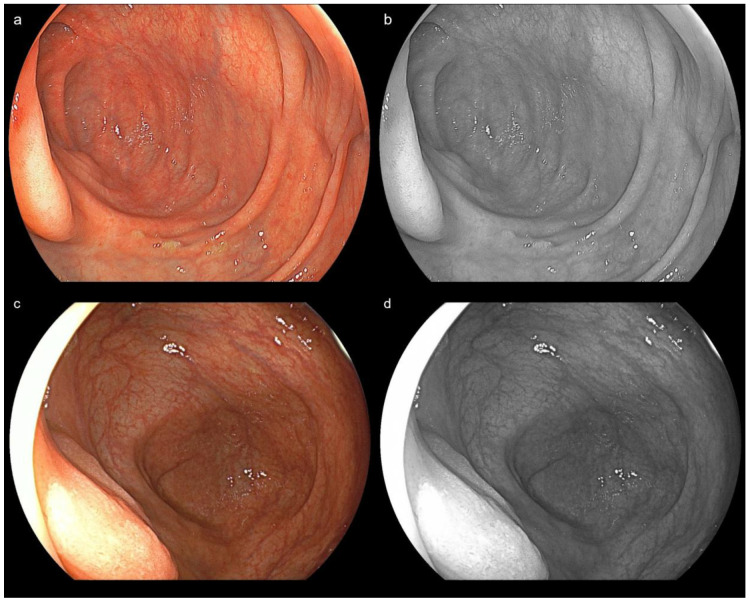
Analysis of the brightness value and proportion of halation in an image performed with an image-analysis software. (**a**) Novel system + Novel scope. Cecal image of WLI. (**b**) The image was arranged by an image-analysis software. Brightness value: 132, halation rate: 0.63%. (**c**) Previous system + previous scope. Cecal image of WLI. (**d**) The image was arranged by an image-analysis software. Brightness value: 115, halation rate: 10.77%.

**Figure 3 diagnostics-15-01569-f003:**
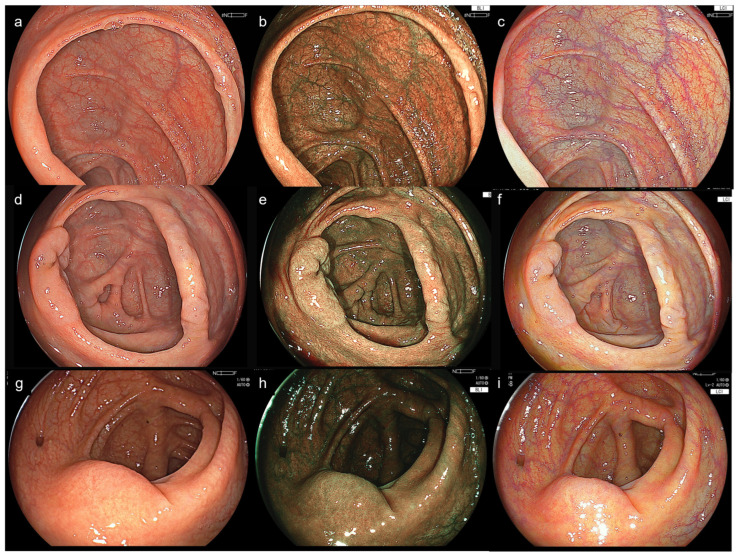
Case presentations of the novel and previous systems for cecal images. (**a**) Novel LED system + Novel scope (Group 3). The endoscopists evaluations of an expert and image software values regarding the brightness/halation were 3/3 and 0.55%/121 for WLI. (**b**) Group 3, 3/3 and 1.45%/114 for BLI. (**c**) Group 3, 3/3 and 0.82%/136 for LCI. (**d**) Novel system + Previous scope (Group 2), 3/3 and 0.83%/117 for WLI. (**e**) Group 2, 3/2 and 1.97%/101 for BLI. (**f**) Group 2, 3/2 and 1.43%/133 for LCI. (**g**) Previous system + Previous scope (Group 1), 3/3 and 0.75%/111 for WLI. (**h**) Group 1, 1/2 and 1.41%/67 for BLI. (**i**) Group 1, 2/1 and 2.47%/123 for LCI.

**Figure 4 diagnostics-15-01569-f004:**
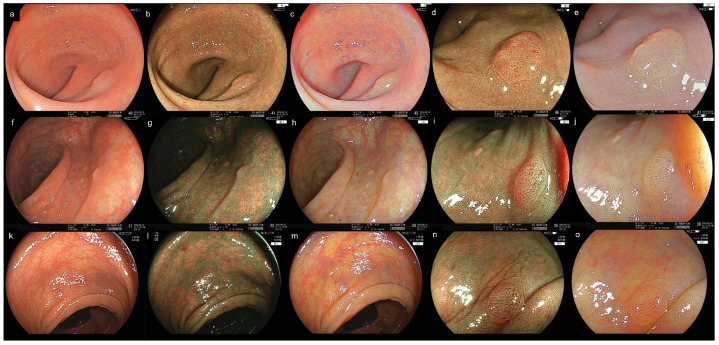
Case presentations of the novel and previous system for lesions images. (**a**) Novel system + Novel scope (Group 3). The endoscopists evaluations of an expert regarding the brightness/halation/visibility were 3/3/2 for WLI. (**b**) Group 3, 3/3/3 for BLI. (**c**) Group 3, 3/3/3 for LCI. (**d**) Group 3, 3/2/3 for BLI magnification. (**e**) Group 3, 3/2/2 for LCI magnification. (**f**) Novel system + Previous scope (Group 2), 3/3/2 for WLI. (**g**) Group 2, 2/3/3 for BLI. (**h**) Group 2, 3/3/2 for LCI. (**i**) Group 2, 3/2/3 for BLI magnification. (**j**) Group 2, 3/3/2 for LCI magnification. (**k**) Previous system + Previous scope (Group 1), 3/1/1 for WLI. (**l**) Group 1, 2/1/2 for BLI. (**m**) Group 1, 3/1/2 for LCI. (**n**) Group 1, 2/1/3 for BLI magnification. (**o**) Group 1, 2/1/2 for LCI magnification.

**Figure 5 diagnostics-15-01569-f005:**
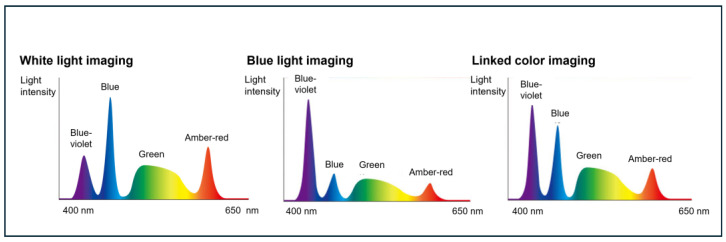
The balance of four color LEDs in the novel system (EP-8000) for WLI, BLI, and LCI.

**Figure 6 diagnostics-15-01569-f006:**
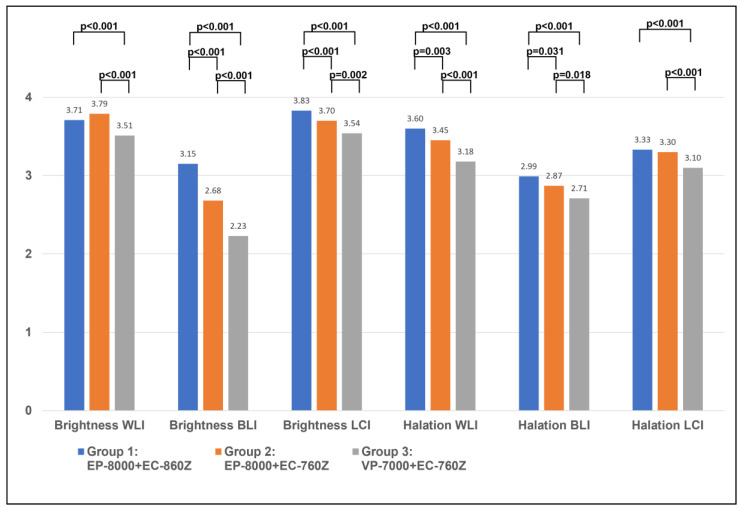
The evaluation by endoscopists of cecal location for WLI, BLI, and LCI in the new system and the previous system. The evaluation of brightness and halation using the 4-point score by endoscopists.

**Figure 7 diagnostics-15-01569-f007:**
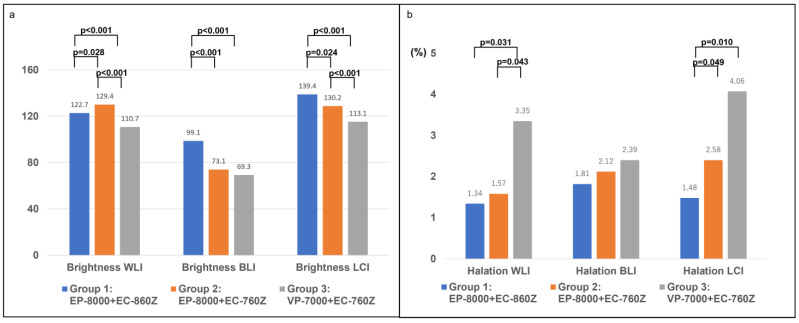
The evaluation by image software for images of cecal location for WLI, BLI, and LCI in the new system and the previous system. (**a**) The evaluation of brightness by image software. (**b**) The evaluation of halation by image software.

**Table 1 diagnostics-15-01569-t001:** Patient and lesion characteristics.

Lesion Number	Group 1 EP-8000+EC-860ZP	Group 2 EP-8000+EC-760ZP	Group 3 VP-7000+EC-760ZP	*p*-Value
Patient number	50	50	50	
Age, mean ± SD (range)	68.2 ± 10.4 (34–86)	66.0 ± 14.4 (20–83)	66.8 ± 11.6 (30–83)	0.706
Sex, male/female, *n* (%)	25/25 (50.0/50.0)	27/23 (54.0/46.0)	32/18 (64.0/36.0)	0.348
Bowel preparation (good/bad)	41/9 (82.0/18.0)	39/11 (78.0/22.0)	43/7 (86.0/14.0)	0.581
Polyp number	56	62	52	
Tumor location, right-sided/left-sided/rectum, *n* (%)	39/11/6 (63.9/14.8/21.3)	49/10/3 (79.0/16.1/4.8)	35/12/5 (67.3/23.1/9.6)	0.603
Morphology type, polypoid/nonpolypoid, *n* (%)	28/28 (50.0/50.0)	23/39 (37.1/62.9)	23/29 (44.2/55.8)	0.366
Tumor size, mm, mean ± SD (range)	4.1 ± 3.5 (2–20)	4.1 ± 3.2 (1–20)	4.5 ± 3.3 (2–18)	0.753
Histopathological diagnosis, SSL/LGA/HGD + T1a, *n* (%)	16/38/2 (28.6/67.9/3.6)	17/43/2 (27.4/69.4/3.2)	13/35/4 (25.0/67.3/7.7)	0.812

SD: standard deviation, right-sided: cecum to transverse colon, left-sided: descending colon to sigmoid colon, SSL: sessile serrated lesions, LGA: low-grade adenoma, HGD: high-grade dysplasia.

**Table 2 diagnostics-15-01569-t002:** The evaluation of brightness, halation, and visibility of lesions for WLI, BLI, and LCI without magnification.

		Group 1 EP-8000+EC-860ZP	Group 2 EP-8000+EC-760ZP	Group 3 VP-7000+EC-760ZP	*p*-Value Group 1 vs. 3	*p*-Value Group 2 vs. 3	*p*-Value Group 1 vs. 2
Brightness Overall, mean ± SD	WLI	3.72 ± 0.50	3.70 ± 0.48	3.51 ± 0.57	<0.001	<0.001	0.336
	BLI	3.51 ± 0.67	3.38 ± 0.71	3.01 ± 0.71	<0.001	<0.001	0.017
	LCI	3.89 ± 0.32	3.79 ± 0.43	3.60 ± 0.56	<0.001	<0.001	0.001
Brightness Experts, mean ± SD	WLI	3.85 ± 0.35	3.82 ± 0.40	3.59 ± 0.56	<0.001	<0.001	0.243
	BLI	3.63 ± 0.60	3.44 ± 0.60	2.99 ± 0.59	<0.001	<0.001	0.007
	LCI	3.94 ± 0.18	3.86 ± 0.33	3.62 ± 0.59	<0.001	<0.001	0.004
Brightness Non-experts, mean ± SD	WLI	3.59 ± 0.59	3.59 ± 0.52	3.44 ± 0.57	0.025	0.019	0.474
	BLI	3.44 ± 0.70	3.38 ± 0.71	3.02 ± 0.80	<0.001	<0.001	0.279
	LCI	3.82 ± 0.40	3.72 ± 0.46	3.62 ± 0.59	<0.001	0.004	0.055
Halation Overall, mean ± SD	WLI	3.42 ± 0.59	3.19 ± 0.71	3.12 ± 0.73	<0.001	0.137	<0.001
	BLI	3.19 ± 0.64	3.03 ± 0.68	2.86 ± 0.73	<0.001	0.004	0.003
	LCI	3.29 ± 0.61	3.12 ± 0.63	3.00 ± 0.65	<0.001	0.026	<0.001
Halation Experts, mean ± SD	WLI	3.58 ± 0.56	3.25 ± 0.75	3.20 ± 0.78	<0.001	0.290	<0.001
	BLI	3.41 ± 0.54	3.21 ± 0.65	2.98 ± 0.76	<0.001	0.006	0.005
	LCI	3.47 ± 0.55	3.17 ± 0.55	3.02 ± 0.70	<0.001	0.052	<0.001
Halation Non-experts, mean ± SD	WLI	3.25 ± 0.58	3.12 ± 0.65	3.02 ± 0.67	0.003	0.149	0.045
	BLI	2.96 ± 0.64	2.83 ± 0.64	2.73 ± 0.68	0.005	0.114	0.068
	LCI	3.12 ± 0.61	3.05 ± 0.57	2.97 ± 0.59	0.032	0.137	0.188
Visibility Overall, mean ± SD	WLI	3.28 ± 0.73	3.01 ± 0.87	3.09 ± 0.82	0.005	0.172	<0.001
	BLI	3.69 ± 0.54	3.46 ± 0.68	3.36 ± 0.69	<0.001	0.069	<0.001
	LCI	3.63 ± 0.59	3.40 ± 0.70	3.31 ± 0.71	<0.001	0.119	<0.001
Visibility Experts, mean ± SD	WLI	3.50 ± 0.65	3.10 ± 0.90	3.08 ± 0.83	<0.001	0.437	<0.001
	BLI	3.83 ± 0.42	3.54 ± 0.62	3.37 ± 0.68	<0.001	0.023	<0.001
	LCI	3.84 ± 0.40	3.52 ± 0.68	3.35 ± 0.65	<0.001	0.029	<0.001
Visibility Non-experts, mean ± SD	WLI	3.05 ± 0.73	2.92 ± 0.81	3.09 ± 0.80	0.684	0.059	0.107
	BLI	3.55 ± 0.61	3.36 ± 0.71	3.34 ± 0.69	0.010	0.429	0.014
	LCI	3.41 ± 0.66	2.26 ± 0.69	3.27 ± 0.75	0.073	0.447	0.043

WLI: white light imaging, BLI: blue light imaging, LCI: linked color imaging, SD: standard deviation.

## Data Availability

The patient data used to support the findings of this study are available from the corresponding author upon request. However, some data is restricted by the institutional review board of the Kyoto Prefectural University of Medicine and the institutional review board of Jichi Medical University.

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
