# Peer review of "Improvement of Colonoscopic Image Quality Using a New LED Endoscopic System with Specialized Noise Reduction"

_diagnostics, 2025, doi:10.3390/diagnostics15121569_

Round 1
Reviewer 1 Report
Comments and Suggestions for Authors
Image quality is a key criterion for all areas of endoscopy.
In this respect, the present study is relevant.
It concerns imaging in the colon by flexible endoscopy and compares the latest generation of a endoscope- manufacturer with high-end-quitetechnology with the previous generation. The changes in technology concern both the illumination modalities (WLI, BLI, LCI) and the software algorithms for image processing in the processor (TNR, EDRIP) before the image appears on the monitor.
This paper is a carefully and well-designed intensive and extensive study to analyze the image quality determined by subjective and objective parameters. The former concern brightness, halation and visibility of lesions by experienced and inexperienced examiners, objective parameters are determined by a software-based evaluation (Photoshop).
The results show a significant superiority of the latest system in all parameters compared to the quite modern previous version.
Critical remarks:
The overwhelming abundance of individual results is presented both in detailed tables and again in the text. From a scientific point of view, the tables would suffice; the text should only refer to special features.
In the introduction and in the chapter on material and methods, the technical background (as far as possible for patent reasons) should be better explained (e.g. LASER system versus LED system, background of LCI? What does TNR include?, How does EDRIP work?
In the discussion chapter, the differentiation from other image optimization methods (e.g. NBI) and their optimizations comes up far too short.
Objectification using a commercial image processing program (Photoshop) may also be questioned for medical purposes. Endoscopic images have their specifics here.
The nomenclature regarding LASER and LED should also be clarified again in the discussion. Comments on how the new system behaves under less than optimal conditions (stool contamination, bleeding and blood residue, fluid reflections, etc.) would also be useful.
From a formal point of view, in addition to the desirable restriction of the results to tables with only a few comments in the text (see above), the presentation of Figure 5 should definitely be improved; this figure cannot be read without a magnifying glass and needs to be printed better and larger.
A list of abbreviations at the beginning of the paper, as is customary internationally, would also be useful.
Author Response
Reviewer 1
Image quality is a key criterion for all areas of endoscopy.
In this respect, the present study is relevant.
It concerns imaging in the colon by flexible endoscopy and compares the latest generation of a endoscope- manufacturer with high-end-quitetechnology with the previous generation. The changes in technology concern both the illumination modalities (WLI, BLI, LCI) and the software algorithms for image processing in the processor (TNR, EDRIP) before the image appears on the monitor.
This paper is a carefully and well-designed intensive and extensive study to analyze the image quality determined by subjective and objective parameters. The former concern brightness, halation and visibility of lesions by experienced and inexperienced examiners, objective parameters are determined by a software-based evaluation (Photoshop).
The results show a significant superiority of the latest system in all parameters compared to the quite modern previous version.
Critical remarks:
The overwhelming abundance of individual results is presented both in detailed tables and again in the text. From a scientific point of view, the tables would suffice; the text should only refer to special features.
Answer: Thank you for your comment. We decreased the amount of the explanation about each table.
In the introduction and in the chapter on material and methods, the technical background (as far as possible for patent reasons) should be better explained (e.g. LASER system versus LED system, background of LCI? What does TNR include?, How does EDRIP work?
Answer: We appreciate your comment. We explained the technical back ground of LASER and LED system, LCI, TNR, and EDRIP as far as possible for patent reasons described below. Additionally, we made a new figure about the new system (Supplemental Figure 2).
Introduction
A laser endoscopic system, LASEREO (VP-4450HD, Fujifilm, Tokyo, Japan), was introduced in 2012, employing two-wavelength lasers (410 nm and 450 nm) for white light imaging (WLI) and blue laser imaging (BLI), which enhance lesion characterization and detection [1–6]. Bandwidths of the two lasers were less than about 2 nm The laser with 450 nm wavelength made phosphor irradiate an illumination similar to that of a xenon lamp which was used in endoscopic system regularly. BLI is a narrow-band observation mode and has two modes such as regular BLI mode for lesion characterization and BLI-bright mode with high illumination for lesion detection. However, BLIs can appear dark, and residual fluid under poor preparation may become reddish. Linked color imaging (LCI) was subsequently developed as a kind of narrow-band light observation mode, providing a brighter field of view and turning fluid yellowish using two lasers, thereby making neoplastic lesions and inflammatory areas appear reddish [7,8]. Multiple studies have demonstrated improved lesion visibility and polyp detection using LCI [9–15]. An LED-based endoscopic system, ELUXEO (VP-7000, Fuji-film) with a dedicated scope (EC-760ZP), was later introduced in 2017 in Western countries and in 2020 in Japan [16,17], utilizing four-color LEDs (blue-violet, blue, green, red). Multi Light technology provides high-quality images with blue light im-aging (It is also called as BLI) as well as LCI for narrow-band light observation.
Method section
The novel LED endoscopic system and magnifying colonoscope used in this study were the EP-8000 and EC-860ZP (Fujifilm Co., Tokyo, Japan). This system utilize different four-color LEDs (blue-violet, blue, green, amber-red) compared to the previous system and incorporates two key features: TNR and E-DRIP (Supplemental Figure 2).
TNR incorporates three noise suppression technologies, each derived from different imaging modalities, to reduce noise in blood vessels and mucosal surface structures, thereby improving visibility compared to conventional systems. These technologies include: 1) Conventional endoscopic noise suppression, which removes sudden noise by comparing consecutive frames. 2) X-ray-based noise suppression, which enhances the clarity of thin blood vessels. 3) Ultrasound-based noise suppression, which improves the contrast of mucosal details.
E-DRIP optimizes tone control, brightens dark areas, and expands the dynamic range. Additionally, its improved exposure control minimizes the occurrence of "crushed blacks" and "clipped highlights." These advancements result in brighter images with reduced halation and enhanced visibility even at a distance (Supplementary video).
This system enables to perform WLI, BLI, and LCI and it also has a new observational mode named as amber-red color imaging for enhancing vessels during endoscopic submucosal dissection.
In the discussion chapter, the differentiation from other image optimization methods (e.g. NBI) and their optimizations comes up far too short.
Answer: We appreciate your comment. We included other image techniques including NBI and TXI from Olympus in the Discussion section describe below, referring some papers.
The Discussion section
There are several specialized observational modes with different mechanisms for improving lesion detection, in addition to BLI and LCI. In narrow-band imaging (NBI), optical filters that allow narrow-band light at wavelengths of 415 and 540 nm to pass through are mechanically inserted between a xenon lamp and an RGB rotary filter [31]. NBI enhances vascular patterns and pit-like structures, and is used for both lesion de-tection and characterization. Similar to BLI, residual fluid appears reddish under NBI.Texture and Color Enhancement Imaging (TXI) is another imaging mode devel-oped to improve lesion visibility by enhancing three image-related factors: texture, brightness, and color [32]. The EVIS X1 (CV-1500; Olympus Co.) is an LED endoscopic system that uses five LED light sources, enabling the use of TXI as well as improved brightness and image resolution in NBI. In TXI, residual fluid appears yellowish, and the endoscopic view remains bright even under poor bowel preparation. Several ran-domized controlled trials (RCTs) have demonstrated the efficacy of NBI and TXI in polyp detection compared to white-light imaging (WLI) [33–36].
Reference
- Sano Y, Muto M, Tajiri H, et al. Optical/digital chromoendoscopy during colonoscopy using narrow-band image system. Dig Endosc 2005;17:S43–8.
- Tamai N, Horiuchi H, Matsui H, et al. Visibility evaluation of colorectal lesion using texture and color enhancement imaging with video. DEN Open 2022;2:e90.
- Antonelli G, Bevivino G, Pecere S, et al. Texture and color enhancement imaging versus high definition white-light endoscopy for detection of colorectal neoplasia: a randomized trial. Endoscopy 2023; 55 :1072-1080.
- Toyoshima N, Mizuguchi Y, Takamaru H, et al. The Efficacy of Texture and Color Enhancement Imaging Observation in the Detection of Colorectal Lesions: A Multicenter, Randomized Controlled Trial (deTXIon Study). Gastroenterology. 2025 Mar 18:S0016-5085(25)00524-4. doi: 10.1053/j.gastro.2025.03.007. Epub ahead of print. PMID: 40113100.
- Yoshida N, Inagaki Y, Inada Y, et al. Additional 30-Second Observation of the Right-Sided Colon for Missed Polyp Detection With Texture and Color Enhancement Imaging Compared with Narrow Band Imaging: A Randomized Trial. Am J Gastro-enterol 2024; 119: 539-546.
- Atkinson NSS, Ket S, Bassett P, et al. Narrow-band imaging for detection of neoplasia at colonoscopy: a meta-analysis of data from individual patients in randomized controlled trials. Gastroenterology 2019:157:462-471.
Objectification using a commercial image processing program (Photoshop) may also be questioned for medical purposes. Endoscopic images have their specifics here.
Answer: We appreciate your comment. We would like to clarify that the use of Photoshop in our study was strictly limited to objective image processing for the evaluation of brightness and halation. No image enhancement or alteration of image appearance was performed. The integrity of the endoscopic findings was fully preserved, and all original image data were retained. We added this explanation in the Method section.
The nomenclature regarding LASER and LED should also be clarified again in the discussion.
Answer: We appreciate your comment. We added the explanation of LASER and LED in the Introduction and Discussion section described below.
The Introduction section
Bandwidths of the two lasers were less than about 2 nm The laser with 450 nm wave-length made phosphor irradiate an illumination similar to that of a xenon lamp which was used in endoscopic system regularly. However, BLI is a narrow-band observation mode and has two mode such as regular BLI mode for lesion characterization and BLI-bright mode with high illumination for lesion detection. However, BLIs can appear dark, and residual fluid under poor preparation may become reddish. Linked color imaging (LCI) was subsequently developed as a kind of narrow-band light observation mode, providing a brighter field of view and turning fluid yellowish using two lasers, thereby making neoplastic lesions and inflammatory areas appear reddish [7,8].
Multi Light technology provides high-quality images with blue light imaging (It is also called as BLI) as well as LCI for narrow-band light observation.
The Discussion section
In the LED system, multilight technology with four LED lights enables to WLI, BLI, and LCI modes. On the other hand, in the laser system, two lasers and phosphor enable to perform WLI, BLI and LCI modes.
Comments on how the new system behaves under less than optimal conditions (stool contamination, bleeding and blood residue, fluid reflections, etc.) would also be useful.
Answer: We appreciate your comment. We added the use of this new system for the comments for some optimal conditions in the Discussion section.
The Discussion section
According to our results, the clinical use of the new LED system—with enhanced brightness and reduced halation—is effective for both routine and magnified observation using WLI, BLI, and LCI. In cases of poor bowel preparation with substantial residual fluid, the increased brightness of WLI and LCI may facilitate improved lesion detection. Furthermore, the brighter BLI provided by the new system enables its use for lesion detection, overcoming the limitation of insufficient brightness seen in previous BLI systems. Further analysis is warranted to validate and clarify these findings.
From a formal point of view, in addition to the desirable restriction of the results to tables with only a few comments in the text (see above), the presentation of Figure 5 should definitely be improved; this figure cannot be read without a magnifying glass and needs to be printed better and larger.
Answer: We appreciate your comment. We deleted the amount of abundant explanation in each table. We also amended Figure 5. Figure 5abc was divided into Figure 5 and Figure 6.
A list of abbreviations at the beginning of the paper, as is customary internationally, would also be useful.
Answer: Thank you for your comment. We made it.
Reviewer 2 Report
Comments and Suggestions for Authors
Dear authors.
Thank you for sharing with us the present manuscript.
I just have two comments:
- First of all, please, could you clarify if this technology also allows narrow band imaging (NBI) in flat lesions?
- Finally, because of the design of your research, you cannot talk about demonstration, just association of two different facts, but it is just an observational study with both retrospective and prospective patients.
Author Response
Reviewer2
Thank you for sharing with us the present manuscript.
I just have two comments:
First of all, please, could you clarify if this technology also allows narrow band imaging (NBI) in flat lesions?
Answer: We included 28 flat lesions with the evaluation of new system in Group 1. We performed subgroup analysis of the comparison for the evaluation of visibility of flat lesions among the three groups as Supplemental Table 3. We added the explanation of the results in the Results section.
The Results section
Additionally, those in non-polypoid lesions were significantly higher in Group 1 than in Group 3 for WLI (p=0.016), BLI (p<0.001), and LCI (p=0.002) and those in polypoid lesions were significantly higher in Group 1 than in Group 3 for BLI (p<0.001) and LCI (p<0.001) (Supplemental Table 3).
Finally, because of the design of your research, you cannot talk about demonstration, just association of two different facts, but it is just an observational study with both retrospective and prospective patients.
Answer: Thank you for your comment. We added the comment of our limitation according to the reviewer’s comment described below.
The Limitation section
Our study only analyzed the improvement of image’s brightness and halation, and lesion’s visibility in the new system. While improved polyp visibility scores may correlate with better polyp detection, actual lesion detection rates in clinical practice is different and can be influenced by many additional factors. Further analysis should be expected to clarify it.